# Improved Anti-Biofilm Effect against the Oral Cariogenic *Streptococcus mutans* by Combined Triclosan/CBD Treatment

**DOI:** 10.3390/biomedicines11020521

**Published:** 2023-02-10

**Authors:** Maayan Avraham, Doron Steinberg, Tamar Barak, Miriam Shalish, Mark Feldman, Ronit Vogt Sionov

**Affiliations:** 1Faculty of Dental Medicine, Ein Kerem Campus, Institute of Biomedical and Oral Research (IBOR), The Hebrew University of Jerusalem, Jerusalem 9112102, Israel; 2Division of Biotechnology, Strauss Campus, Hadassah Academic College, Jerusalem 9514223, Israel; 3Hadassah Medical Center, Department of Orthodontics, Faculty of Dental Medicine, The Hebrew University of Jerusalem, Jerusalem 9112102, Israel

**Keywords:** anti-bacterial, anti-biofilm, cannabidiol, dental caries, *Streptococcus mutans*, triclosan

## Abstract

*Streptococcus mutans* is a Gram-positive bacterium highly associated with dental caries, and it has a strong biofilm-forming ability, especially in a sugar-rich environment. Many strategies have been undertaken to prevent dental caries by targeting these bacteria. Recently, we observed that a sustained-release varnish containing triclosan and cannabidiol (CBD) was more efficient than each compound alone in preventing biofilm formation by the fungus *Candida albicans*, which is frequently involved in oral infections together with *S. mutans*. It was therefore inquiring to study the effect of this drug combination on *S. mutans*. We observed that the combined treatment of triclosan and CBD had stronger anti-bacterial and anti-biofilm activity than each compound alone, thus enabling the use of lower concentrations of each drug to achieve the desired effect. The combined drug treatment led to an increase in the SYTO 9^low^, propidium iodide (PI)^high^ bacterial population as analyzed by flow cytometry, indicative for bacteria with disrupted membrane. Both triclosan and CBD induced membrane hyperpolarization, although there was no additive effect on this parameter. HR-SEM images of CBD-treated bacteria show the appearance of elongated and swollen bacteria with several irregular septa structures, and upon combined treatment with triclosan, the bacteria took on a swollen ellipse and sometimes oval morphology. Increased biofilm formation was observed at sub-MIC concentrations of each compound alone, while combining the drugs at these sub-MIC concentrations, the biofilm formation was prevented. The inhibition of biofilm formation was confirmed by CV biomass staining, MTT metabolic activity, HR-SEM and live/dead together with exopolysaccharide (EPS) staining visualized by spinning disk confocal microscopy. Importantly, the concentrations required for the anti-bacterial and anti-biofilm activities toward *S. mutans* were non-toxic to the normal Vero epithelial cells. In conclusion, the data obtained in this study propose a beneficial role of combined triclosan/CBD treatment for potential protection against dental caries.

## 1. Introduction

Biofilm formation by cariogenic bacteria on enamel contributes to the sustained dissolution and loss of minerals, ultimately resulting in tooth decay [1]. *Streptococcus mutans* is a common oral cariogenic bacteria characterized by strong biofilm formation ability, acidogenicity (the ability to produce acids such as lactic acid upon fermentation) and aciduricity (the ability to survive under acidic conditions) [2]. The production of acidic metabolites leads to the acidic destruction and demineralization of the tooth enamel, which is manifested by dental caries. *S. mutans* is a Gram-positive facultatively anaerobic coccus that produces exopolysaccharides (EPSs) from sucrose catalyzed mainly by glucosyltransferases (Gtfs) and to a lesser extent by fructosyltransferase (Ftf) [3]. The EPSs stick to both biotic and abiotic surfaces, assisting the adherence and accumulation of bacteria, thus initiating biofilm formation. The resulting biofilm usually contains several different species of bacteria and fungi encapsulated in EPS, various salivary components, bacterial by-products, and food remnants [4]. Within the biofilm, the microbes are protected from the harsh surroundings and are less susceptible to anti-microbial agents and the immune defense mechanisms [5].

Common approaches to prevent microbial biofilm formation include mechanical brushing and flossing combined with toothpastes and mouthwashes containing antiseptic compounds such as: fluorides, chlorhexidine, triclosan, quaternary ammonium salts and/or herbal extracts. However, some of these compounds are toxic to the surrounding tissues when overused. For example, chlorhexidine can cause tooth discoloration and calcium buildup [6]. Upon severe gingivitis and periodontitis, antibiotics are typically applied to reduce the bacterial burden. However, antibiotics can only be used for a limited period due to the risk of developing drug resistance and distorting the normal microbiome.

Many efforts have been made to find novel approaches to prevent biofilm formation. These include natural products such as polyphenolic flavonoids derived from green tea and other plants (e.g., [7,8,9,10]) and essential herb oils (e.g., [11]). However, many of these compounds had to be applied at high concentrations (in the range of mg/mL) to achieve the desired biological effects.

We recently observed that the non-psychotropic cannabidiol (CBD) derived from the *Cannabis sativa* L. plant is efficient in inhibiting the growth of *S. mutans* as well as in preventing biofilm formation at non-cytotoxic concentrations to mammalian epithelial cells [12]. CBD was also able to reduce the bacterial content of dental plaques obtained from human adults and was found to be better than Colgate, Oral B and Cannabite F toothpastes [13]. Another study incorporated CBD in a mouthwash and observed that this formulation caused bacterial clearance in a disk diffusion assay using bacteria isolated from dental plaques of human adults [14]. Other researchers have also observed a beneficial anti-bacterial action of CBD on various Gram-positive bacteria including both antibiotic-sensitive and antibiotic-resistant *Staphylococcus aureus* strains [15,16,17]. Its ability to also act on drug-resistant bacteria suggests that its action mechanism is not influenced by the drug-resistance mechanisms, which is a major advantage in the drug-resistance era. Blaskovich et al. [17] observed that repeated exposure of *S. aureus* to CBD did not induce resistance to this anti-bacterial compound. CBD has both bacteriostatic and bactericidal effects depending on the concentration used [12,15,16,17]. The underlying action mechanism is still not fully understood, and currently, it is thought to be due to a combination of a membrane effect and an anti-metabolic effect [17,18]. In *S. aureus*, CBD causes membrane depolarization [17], while in *S. mutans*, it causes a transient membrane hyperpolarization [12] indicating species-specific effects. A docking study predicted that CBD might bind to the PBP2a variant of *S. aureus* involved in cell wall synthesis [19].

In contrast to the endocannabinoids anandamide and AraS that have been shown to synergize with various antibiotics and even sensitize multidrug-resistant *S. aureus* (MRSA/MDRSA) to antibiotics [20,21,22], most studies have not been able to prove a synergistic effect between CBD and antibiotics (e.g., the β-lactams ceftazidime, cefepime, aztreonam and meropenem, and the glycopeptide teicoplanin) [17]. There are however some few exceptions where the combination of CBD with the anti-microbial peptide polymyxin B could sensitize several Gram-negative bacteria to CBD [18,23], and the combined treatment of CBD with bacitracin had synergistic anti-bacterial effects against the Gram-positive *S. aureus*, *Listeria monocytogenes* and *Enterococcus faecalis* bacteria [24].

It is worth mentioning that CBD has strong anti-inflammatory activities with beneficial effects on various autoimmune diseases including diabetes, rheumatoid arthritis, and multiple sclerosis, as well as on inflammatory conditions such as inflammatory bowel disease [25,26,27,28]. It is thus likely that CBD might also have beneficial effects on gingivitis, periodontitis and other oral inflammatory processes.

The observation that CBD and triclosan, when incorporated in a sustained-release device, had synergistic activity against the fungus *Candida albicans* [29], led us to test whether these two compounds might also act together on *S. mutans*. Here, we document that when combined, these compounds have a specific anti-biofilm activity against *S. mutans*, distinct from the anti-bacterial activity. These findings suggest that it might be beneficial to explore the use of these two compounds together in mouthwashes, at least in adults, and especially in caries-prone people.

## 2. Materials and Methods

### 2.1. Materials

CBD (purity > 99%) was obtained from NC Labs (Prague, Czech Republic) and dissolved at 10 mg/mL in ethanol. Triclosan was obtained from Sigma (St. Louis, MO, USA) and dissolved at 20 mg/mL in ethanol.

### 2.2. Bacteria Strain and Cultivation

*S. mutans* UA159 was cultivated in brain–heart infusion (BHI) broth (Acumedia, Lansing, MI, USA) [30]. The day before the experiment, 100 µL of a frozen (−80 °C) stock was inoculated in 10 mL BHI followed by incubation at 37 °C in a humidified incubator in the presence of 5% CO_2_ for 20 h. The chain formation and coccoid-to-ellipsoid morphology of the *S. mutans* culture were verified under light microscopy (Axio Lab.A1, Carl Zeiss GmbH, Jena, Germany) using the ×100 lens, and the optical density at 600 nm determined by Ultraspec 10 spectrophotometer (Amersham Biosciences, Amersham, UK), usually being in the range of 1.2–1.3. The purity of the culture was tested by seeding the bacteria on BHI-agar plates where they formed distinctive minute colonies that are strongly attached to the agar.

### 2.3. Planktonic Growth

To study the effect of the compounds on bacterial growth, the overnight culture was diluted to an OD_600 nm_ of 0.1 in fresh BHI and incubated in the absence or presence of various concentrations of each compound alone or in combination in 200 µL BHI in 96 flat-bottomed tissue culture plates (Corning Incorporated, Kennebunk, ME, USA). After a 24 h incubation at 37 °C in the presence of 5% CO_2_, the bacterial growth was determined by measuring the optical density at 600 nm using the M200 Infinite plate reader (Tecan Trading AG, Männedorf, Switzerland) [30]. Control samples got the same concentrations of ethanol as the treated samples. In parallel, untreated bacteria were incubated in BHI. The presence of ethanol at the concentrations used (0.0156–1%) had no effect on bacterial growth. BHI with the compounds in the absence of bacteria was used as background read. The percentage viability was determined by the following formula:(1)$$\%\;\mathrm{Viability}=\frac{\left({\mathrm{OD}}_{600\mathrm{nm}}\;\mathrm{of}\;\mathrm{treated}\;\mathrm{sample}-{\mathrm{OD}}_{600\mathrm{nm}\;}\mathrm{of}\;\mathrm{background}\right)}{\left({\mathrm{OD}}_{600\mathrm{nm}}\;\mathrm{of}\;\mathrm{control}\;\mathrm{sample}-{\mathrm{OD}}_{600\mathrm{nm}}\;\mathrm{of}\;\mathrm{background}\right)}\times 100\%$$

The MIC was defined as the lowest concentration that inhibited visible growth of the bacteria (ISO 17,025 standard). To calculate whether the combined treatment is synergistic, additive, or antagonistic, the fractional inhibitory concentration index (FICI) value was determined by the following formula [31]:(2)$$\mathrm{FICI}=\frac{{\mathrm{MIC}}_{A}\;\mathrm{in}\;\mathrm{combination}}{{\mathrm{MIC}}_{A}\;\mathrm{alone}}+\frac{{\mathrm{MIC}}_{B}\;\mathrm{in}\;\mathrm{combination}}{{\mathrm{MIC}}_{B}\;\mathrm{alone}}$$
where MIC_A_ is the MIC of compound A (in our case CBD), and MIC_B_ is the MIC of compound B (in our case triclosan). A synergistic effect is considered when the FICI value is lower than 0.5. FICI values between 0.5–1 is considered an additive effect. According to the formula, the reduction in the MIC values in combination need to be reduced more than 4-fold for each compound in order to obtain a FICI value lower than 0.5.

### 2.4. Biofilm Formation

To study the effect of the compounds on biofilm formation, the overnight *S. mutans* culture was diluted to an OD_600 nm_ of 0.1 in BHI supplemented with 2% sucrose (BHIS) and incubated with the different drug combinations for 24 h [32]. For the MTT metabolic assay (Section 2.5) and CV staining method (Section 2.6), the bacteria were incubated in a total volume of 200 µL in 96-flat-bottomed tissue culture plates (Corning). For morphological analysis on HR-SEM (Section 2.7), the bacteria were incubated in a total volume of 1 mL in 24-flat-bottomed tissue culture plates (Corning) containing glass slide pieces at the bottom. For SDCM (Section 2.8), the bacteria were incubated in a total volume of 1 mL in 24-flat-bottomed tissue culture plates (Corning). At the end of incubation, the biofilms were washed twice with PBS before being processed for the specified assays.

### 2.5. MTT Metabolic Assay

The biofilms were exposed to 0.5 mg/mL MTT (3-(4,5-Dimethyl-2-thiazolyl)-2,5-diphenyl-2H-tetrazolium bromide; Sigma, St. Louis, MO, USA) for 1 h at 37 °C, followed by washings with PBS [12]. The amount of formazan formed in the biofilms was measured in an M200 Infinite plate reader (Tecan Trading AG, Männedorf, Switzerland) at an optical density of 570 nm. The percentage of metabolically active bacteria was quantified by the following formula:(3)$$\%\;\mathrm{Metabolic}\;\mathrm{activity}=\frac{\left({\mathrm{OD}}_{570\mathrm{nm}}\;\mathrm{of}\;\mathrm{treated}\;\mathrm{sample}-{\mathrm{OD}}_{570\mathrm{nm}\;}\mathrm{of}\;\mathrm{background}\right)}{\left({\mathrm{OD}}_{570\mathrm{nm}}\;\mathrm{of}\;\mathrm{control}\;\mathrm{sample}-{\mathrm{OD}}_{570\mathrm{nm}}\;\mathrm{of}\;\mathrm{background}\right)}\times 100\%$$

### 2.6. CV Staining

The biofilms were stained for 20 min with 0.1% crystal violet solution that was prepared from a 0.4% Gram’s crystal violet stain solution (Merck, EMD Millipore Corporation, Billerica, MA, USA) diluted in DDW. The stained biofilms were washed several times in DDW, and the intensity of the CV staining was measured in an M200 Infinite plate reader (Tecan Trading AG, Männedorf, Switzerland) at an optical density of 595 nm. The percentage biofilm biomass was quantified by the following formula:(4)$$\%\;\mathrm{Biofilm}\;\mathrm{Biomass}=\frac{\left({\mathrm{OD}}_{595\mathrm{nm}}\;\mathrm{of}\;\mathrm{treated}\;\mathrm{sample}-{\mathrm{OD}}_{595\mathrm{nm}\;}\mathrm{of}\;\mathrm{background}\right)}{\left({\mathrm{OD}}_{595\mathrm{nm}}\;\mathrm{of}\;\mathrm{control}\;\mathrm{sample}-{\mathrm{OD}}_{595\mathrm{nm}}\;\mathrm{of}\;\mathrm{background}\right)}\times 100\%$$

### 2.7. High-Resolution Scanning Electron Microscopy (HR-SEM) Imaging

For studying the effect of the compounds on the bacterial morphology, *S. mutans* was incubated under planktonic growth conditions in the absence or presence of the compounds for 2 h, followed by two washes in PBS, a 2 h fixation in 4% glutaraldehyde solution in DDW (diluted from a 25% electron microscope grade glutaraldehyde solution; Electron Microscopy Sciences, Hatfield, PA, USA), two DDW washes, and dried on slide glass pieces. For biofilm samples, the bacteria were cultivated in BHIS in the absence or presence of the compounds for 24 h on slide glass pieces, washed twice in DDW, fixed in 4% glutaraldehyde solution in DDW for 2 h, washed again twice in DDW, and let dry.

The samples were then mounted on a metal stub and sputter coated with iridium and visualized by a HR-SEM (Magellan XHR 400L, FEI Company, Hillsboro, OR, USA). The length of the bacteria was measured at the ×20,000 magnification using the ImageJ software [20]. More than 200 bacteria were measured for each sample.

### 2.8. Scanning Disk Confocal Microscopy (SDCM) Imaging

Biofilms that have been formed in the absence or presence of the compounds after a 24 h incubation were stained with 3.3 µM SYTO 9 (Invitrogen, Life Technologies, Carlsbad, CA, USA), 2 µg/mL propidium iodide (PI; Sigma) and 10 µg/mL AlexaFluor^647^-conjugated Dextran 10,000 (Molecular Probes Inc., Eugene, OR, USA) for 20 min, followed by washes in PBS, fixation in 4% formaldehyde in PBS (diluted from a 32% paraformaldehyde electron microscope grade solution (Electron Microscopy Sciences, Hatfield, PA, USA) in sterile DDW), washes in DDW, and mounting in 50% glycerol in DDW [12]. SYTO 9 penetrates both live and dead bacteria where it binds to nucleic acid and emits green fluorescence upon excitation [33]. PI can only penetrate the bacteria when the membrane has been perforated and thus stains only dead bacteria [33]. However, PI might also stain extracellular DNA that has been released from dying bacteria and entrapped in the biofilm [34]. Dextran binds to EPS and can thus be used to quantify the EPS content of the biofilm.

The samples were visualized by Scanning disk confocal microscope (Nikon Corporation, Tokyo, Japan), using the 50 µm pinhole and 2 × 2 binning function in the NIS element software. Images were taken at 2.5 µm intervals, using the excitation laser 488 nm for SYTO 9, the excitation laser 561 nm for PI, and the excitation laser 640 nm for AlexaFluor^647^. Upon excitation, SYTO 9 emits green fluorescence, PI emits red fluorescence and AlexaFluor^647^ emits a far-red fluorescence, which is presented in our images in a blue color. The fluorescence intensity of each layer of each sample was quantified by the NIS element software by keeping the parameters the same for all samples.

### 2.9. Flow Cytometry

*S. mutans* that has been exposed to the compounds or control bacteria were washed in PBS and incubated with various fluorescent dyes according to the aim of the experiment before being analyzed on flow cytometer. A 20 min staining solution of 3.3 µM SYTO 9 together with 2 µg/mL PI was used to analyze bacteria with disrupted membrane [35]. This staining was performed at room temperature. For determining the membrane potential, the bacteria were exposed to 15 μM 3,3′-diethyloxacarbocyanine iodide (DiOC2(3)) solution in PBS for 30 min at room temperature prior to flow cytometry [20]. For determining the membrane and DNA content, the bacteria were exposed to 10 µg/mL Nile red (APExBIO, Houston, TX, USA) and 1 µg/mL DAPI (Sigma) for 30 min at 37 °C [20]. The following excitation/emission parameters were used: 488 nm/531 nm for SYTO 9 and the green fluorescence of DiOC2(3); 488 nm/620 nm for PI and the red fluorescence of DiOC2(3); 561 nm/635 nm for Nile red and 355 nm/450 nm for DAPI. The samples were analyzed by the LSR-Fortessa flow cytometry instrument (BD Biosciences, San Jose, CA, USA), and 50,000 events were collected using the BD FACSDiva software 6.0. The FCS Express 7 software was used for analyzing the data.

### 2.10. Biocompatibility Assay on Vero Epithelial Cells

Vero kidney epithelial cells (ATCC CCL-81) kindly provided by Dr. Alexander Rouvinski, The Hebrew University of Jerusalem, Israel, were seeded at 4 × 10^5^ cells per well in a 96-well flat-bottomed tissue culture plate (Corning) in 200 µL DMEM (Sigma) supplemented with 8% heat-inactivated fetal calf serum (Sigma), 2 mM L-glutamine, 1 mM sodium pyruvate, 100 U/mL penicillin and 0.1 mg/mL streptomycin (Biological Industries, Beth HaEmek, Israel) and incubated at 37 °C in the presence of 5% CO_2_. On the following day, when a cell monolayer was formed, the medium was exchanged with 200 µL fresh medium containing the indicated concentrations of triclosan and/or CBD followed by a 24 h incubation. At the end of incubation, the cell morphology was inspected under a light microscope, and the cells were either stained with 0.1% CV solution for 20 min at room temperature or MTT was added to the medium at a final concentration of 0.5 mg/mL followed by a 30 min incubation at 37 °C. The cells were washed in PBS and the absorbance at 595 and 570 nm was measured for the CV-stained and MTT-exposed cells, respectively, using the M200 infinite Tecan plate reader.

### 2.11. Statistical Analyses

The experiments were performed in triplicates and statistical significance was determined by Student’s *t* test as calculated by the Microsoft Excel software. Representative experiments are presented. A *p* value below 0.05 was considered statistically significant.

## 3. Results

### 3.1. The Anti-Bacterial and Anti-Biofilm Activities of CBD and Triclosan Alone or in Combination

Both CBD and triclosan have anti-bacterial and anti-biofilm effects against *S. mutans* as shown in Figure 1, with a minimum inhibitory concentration (MIC) and minimum biofilm inhibitory concentration (MBIC) of 5 µg/mL for CBD and 10–20 µg/mL for triclosan, which confirm previous studies [12,36]. As single agents, the MIC and MBIC values were similar for the given compound, and there was even a small increase in the metabolic activity of the biofilms at low sub-MIC concentrations (Figure 1; *p* < 0.05 for 0.625 µg/mL CBD and 2.5 µg/mL triclosan), a phenomenon that has also been observed for several anti-microbial compounds when used at sub-MIC concentrations [37,38]. When used as single agents, most of the anti-biofilm activity seems to be contributed by the anti-bacterial activity, although we cannot exclude a direct anti-biofilm activity.

We were interested in studying whether the combined treatment of CBD and triclosan could increase the anti-bacterial and anti-biofilm actions against *S. mutans*. To this end, *S. mutans* was treated with different concentrations of the two compounds under planktonic growth or biofilm formation conditions, and the resulting bacterial growth and biofilms were studied after a 24 h incubation. Under these settings, the MIC_CBD_ was found to be 2.5 µg/mL when combined with 5 µg/mL triclosan (Figure 2A,B), providing a fractional inhibitory concentration index (FICI) value of 1 (MIC_CBD_ reduced from 5 to 2.5 µg/mL, and MIC_triclosan_ from 10 to 5 µg/mL), indicating an additive effect rather than a synergistic effect. Interestingly, the biofilm formation was prevented by 2.5 µg/mL CBD when combined with 2.5 µg/mL triclosan as determined by the MTT metabolic activity (Figure 2C,D), which is a lower concentration of triclosan than what is required for achieving the anti-bacterial effect (Figure 2A,B). This finding is especially noteworthy considering the higher concentration of triclosan required to affect biofilm formation in comparison to the free-floating planktonic growth when used as a single agent (Figure 1D,F versus Figure 1B; at 5 µg/mL triclosan, there was a 50% reduction in viability versus no reduction in biofilm mass). Crystal violet staining that dyes the extracellular matrix as well as both live and dead bacteria showed a slightly higher biofilm mass (Figure 2E,F) when compared to the data obtained by the MTT metabolic assay (Figure 2C,D). Nevertheless, the anti-biofilm activity of the combined CBD/triclosan treatment can clearly be recognized by this assay (Figure 2E,F). The FICI value for biofilm was 0.75 (MBIC_CBD_ reduced from 5 to 2.5 µg/mL, and MBIC_triclosan_ from 10 to 2.5 µg/mL), again pointing to an additive effect.

### 3.2. The Effect of Combined CBD/Triclosan Treatment on Membrane Permeability

The combined staining of bacteria with the nucleic acid dyes SYTO 9 and propidium iodide (PI) might provide information on the membrane permeability of the bacteria [35]. SYTO 9 is a neutral molecule that can penetrate both live and dead bacteria, while PI is positively charged and penetrates only disrupted membranes. When performing flow cytometry of SYTO 9/PI-stained bacteria, bacteria with disrupted membrane appear as PI^high^ SYTO9^low^ [35] as indicated by an arrow in Figure 3B–H and summarized in Figure 3J. Both triclosan alone (Figure 3B,C) and CBD alone (Figure 3D–F) after a 1 h incubation with the bacteria induced membrane permeabilization, which was further increased upon combined treatment (Figure 3G–J).

### 3.3. The Effect of Combined CBD/Triclosan Treatment on Nile Red Membrane Staining

Nile red when integrated in the membrane emits red fluorescence than can be used as a measure to quantify the bacterial membrane [39]. Nile red was used in combination with DAPI that freely diffuses into the bacteria and stains DNA [39]. We observed that both CBD and triclosan diminished DAPI staining, and each compound reduced Nile red staining in a dose-dependent manner after 1 h incubation (Figure 4). The combined treatment did not lead to a further reduction in either of the dyes (Figure 4). The relative reduction in Nile red staining was similar to the reduction in DAPI staining except for the high MIC concentrations of CBD (5 µg/mL) and triclosan (10 µg/mL), where the reduction in Nile red fluorescence intensity was greater than that of DAPI (Figure 4G). It is therefore likely that the reduced staining by the compounds at the sub-MIC concentrations is caused by modulating the activity of efflux pumps rather than an effect on the DNA and membrane content. Both DAPI and Nile red are substrates of efflux pumps [20,40,41], and several efflux pumps have been characterized in *S. mutans* [42].

### 3.4. Both CBD and Triclosan Induces Membrane Hyperpolarization

The bacterial membrane potential was studied after a 1 h incubation with triclosan and CBD alone or in combination using the DiOC2(3) potentiometric dye. CBD induced a dose-dependent hyperpolarization (Figure 5A,B), while membrane hyperpolarization was only observed with 2.5 μg/mL triclosan at this time point (Figure 5C,D). Membrane hyperpolarization was also observed when both compounds were used together (Figure 5E–G; and Appendix A). After a 3 h incubation with each compound alone, the bacterial membrane potential had returned to that of the control bacteria (Appendix A), while it was still hyperpolarized in bacteria exposed to the combined triclosan/CBD treatment (Appendix A).

### 3.5. Morphological Alterations of CBD- and/or Triclosan-Treated S. mutans as Revealed by HR-SEM

To study the short-term effect of triclosan and/or CBD on the morphology of the bacteria, these were exposed to various combinations of the compounds for 2 h, followed by imaging by high-resolution scanning electron microscopy (HR-SEM) (Figure 6). Already at 1.25 µg/mL CBD, some swollen and elongated bacteria can be observed (Figure 6B and Appendix A). The average length was increased from 0.85 ± 0.16 µm for control bacteria to 1.03 ± 0.27 µm for those treated with 1.25 μg/mL CBD (Figure 7). Although the average length was not further increased by higher concentrations of CBD (0.99 ± 0.29 µm for 5 µg/mL CBD; Figure 7), there were many bacteria with altered morphology displaying more electron-lucent rings around the bacteria at 5 µg/mL CBD (Figure 6D and Appendix A), which seem to be multiple irregular septa [43]. Usually, two of these ring structures can be observed in the center of the control bacteria (Figure 6A and Appendix A) which appear as the classical ovococci typical for *S. mutans* [43]. The triclosan-treated (5 µg/mL) bacteria also show several elongated bacteria (Figure 6E and Appendix A) with an average length of 1.04 ± 0.31 µm (Figure 7). When combining 5 µg/mL triclosan with 2.5 µg/mL CBD, the average length increased to 1.19 ± 0.33 µm (Figure 6G and Figure 7, Appendix A). When 5 µg/mL triclosan was combined with 5 µg/mL CBD, the bacteria appeared in ellipse form with signs of swollenness (Figure 6H and Appendix A).

### 3.6. HR-SEM Images of CBD- and/or Triclosan-Treated S. mutans Biofilms

We next studied the morphology of the bacteria in the biofilms formed following exposure to triclosan and/or CBD (Figure 8). The control samples show full biofilm coverage of the surface with bacteria enwrapped in an extracellular matrix, and clusters of bacteria are seen in the “hills” and “valleys” of the biofilm (Figure 8A). A similar biofilm morphology was observed in the biofilms formed in the presence of 1.25 µg/mL CBD (Figure 8B). Biofilms formed in the presence of 2.5 µg/mL CBD still showed clusters of bacteria, but there was significantly more extracellular matrix around these treated bacteria (Figure 8C) than in the control samples (Figure 8A). The bacteria in the biofilms that had formed in the presence of 5 µg/mL triclosan showed a more organized, often curved, structure and lacked the classical bacterial clusters as seen in the control samples (Figure 8D versus Figure 8A). The bacteria become more rounded, and still, extracellular matrix could be clearly discerned (Figure 8D). These features become even more pronounced when biofilms were formed in the presence of both 5 µg/mL triclosan and 1.25 µg/mL CBD (Figure 8E). However, when combining 5 µg/mL triclosan and 2.5 µg/mL CBD, only scattered bacteria could be observed (Figure 8F), indicating that this condition showed strong anti-biofilm activity. Parallel biofilm samples stained with CV showed similar reduction in biofilm formation with the combined triclosan/CBD treatment (Appendix A), which also confirms the data presented in Figure 2.

### 3.7. Dead/Live Staining of CBD- and/or Triclosan-Treated S. mutans Biofilms

To know whether the bacteria in the treated biofilms are viable, we performed SYTO 9/PI live dead staining together with fluorescent Dextran 10,000 that binds to EPS followed by spinning disk confocal microscopy (SDCM) (Figure 9 and Appendix A). Biofilms formed in the presence of CBD alone (1.25 or 2.5 µg/mL) or triclosan alone (2.5 or 5 µg/mL) showed strong SYTO 9 and EPS staining with scattered PI staining (Figure 9 and Figure 10, Appendix A). There was a slight increase in PI fluorescence intensity in comparison to SYTO 9 staining when the biofilms were formed in the presence of 5 µg/mL triclosan (Appendix A). Importantly, when triclosan was combined with CBD (2.5 µg/mL triclosan with 2.5 µg/mL CBD or 5 µg/mL triclosan with 1.25 µg/mL CBD), there was almost no SYTO 9 staining, while the few bacteria attached to the surface showed strong PI staining surrounded by EPS staining (Figure 9 and Appendix A). Quantification of the relative PI staining to SYTO 9 indicated that nearly all the bacteria were dead upon combined triclosan/CBD treatment (Figure 10 and Appendix A).

### 3.8. Cytotoxicity Assay on Vero Cells

It was important to test the biocompatibility of the combined triclosan/CBD treatment on normal epithelial cells. To this end, we used the Vero cells that have become the Gold standard for this purpose according to ISO 10993-5 (2009) recommendations [44]. The Vero cells in monolayer were exposed to various concentrations of each compound alone or together, and the viability of the cells after a 24 h incubation was inspected visually by light microscopy and analyzed by CV staining and MTT metabolic assay. Inspection under light microscopy showed that CBD was cytotoxic at 25 μg/mL, while triclosan was cytotoxic at 50 μg/mL. This was confirmed by CV staining and MTT assay with no significant staining at these concentrations (Figure 11). A 33–35% reduction of the CV-stained cell mass was observed with 12.5 μg/mL CBD, despite normal cell morphology, which might be an indication for reduced cell growth. CBD reduced the metabolic activity of Vero cells by 25–50% at 1.56–12.5 μg/mL (Figure 11B), which is reflected in a lower MTT/CV index. The combined treatment with 5 or 10 μg/mL triclosan did not increase the susceptibility of Vero cells to CBD (Figure 11A,B). Thus, the concentrations required for achieving the anti-biofilm activity of triclosan with CBD is far below the cytotoxic concentrations.

## 4. Discussion

Maintaining a healthy oral cavity is not only important for preventing dental caries and oral inflammatory diseases such as gingivitis and periodontitis, but it also has an impact on the general health of the patients since the oral microbiota influences the physical state of humans and can induce systemic diseases such as diabetes, obesity, rheumatoid arthritis and even cancer [45,46]. There is also a relationship between oral and gut microbiota with its implications [45,47]. Keeping good oral hygiene often involves the use of mouthwashes containing antiseptic compounds such as chlorhexidine, triclosan or cetylpyridinium chloride that have broad-acting anti-bacterial effects [48]. Many studies have searched for natural compounds that can have beneficial anti-cariogenic effects, including components of green tea such as epigallocatechin gallate (EGCG) [49]. EGCG has the disadvantage of causing dehydration, and relatively high concentrations (in the range of 1–10 mg/mL) are required to achieve the desired effects [10]. Other natural compounds that have been tested for their anti-bacterial effects include curcumin, kaempferol, quercetin and resveratrol [50]. Also in these cases, high concentrations are required for the anti-bacterial effects [51,52].

During the last several years, non-psychotropic compounds such as CBD and CBG from the *Cannabis sativa* L. plant have attracted attention for the treatment of various diseases due to their neuroprotective and anti-inflammatory properties [27,53,54]. These compounds have been shown to relieve autoimmune diseases such as diabetes, rheumatoid arthritis and multiple sclerosis, as well as various inflammatory conditions [27]. Additionally, both CBD and CBG have been shown to exert anti-bacterial and anti-biofilm activities against Gram-positive bacteria including drug-sensitive and drug-resistant *Staphylococcus aureus* as well as the oral cariogenic *Streptococcus mutans* bacterium at concentrations as low as 1–10 µg/mL [12,15,16,17,30,32]. Moreover, they may have an anti-bacterial effect against certain Gram-negative bacteria when combined with polymyxin B or colistin (polymyxin E) [18,23,55]. Their ability to act on drug-resistant bacteria [16,17] is an advantage since their action mechanism is not influenced by drug-resistant mechanisms, and as such, they will have a broader anti-bacterial spectrum. CBG has also been shown to inhibit quorum sensing in the Gram-negative marine *Vibrio harveyi* [56]. This is an important trait since quorum sensing is involved in the regulation of both antibiotic resistance and biofilm formation [57]. Only at high concentrations of CBD (above 25 µg/mL), which is cytotoxic to normal epithelial cells, could significant anti-biofilm and anti-fungal activities against the *Candida albicans* fungus be observed [58]. However, when incorporated into a sustained release varnish, CBD in combination with triclosan at sub-cytotoxic concentrations had a synergistic effect against *C. albicans* [29]. *C. albicans* is an opportunistic pathogen involved in various infectious diseases including that of the oral cavity [59] and might form dual-species biofilms with *S. mutans* [60].

In light of these findings, we were interested to study whether the combined treatment of CBD with the antiseptic compound triclosan shows improved anti-bacterial and anti-biofilm activity against *S. mutans*. Indeed, our study demonstrates that using a combination of sub-MIC concentrations of these two compounds resulted in reduced viability of the bacteria and prevented biofilm formation. The meaning of this is that lower concentrations of the compounds can be used when combined together, which is of great advantage. Importantly, the effective concentrations of this drug combination were not cytotoxic toward normal epithelial cells as demonstrated using Vero epithelial cells.

Our study also showed that lower concentrations of the two drugs were required to achieve the anti-biofilm activity in comparison to those needed for the anti-bacterial activity. This is in contrast to the use of the single agents where MBIC values were similar to the MIC values. This finding indicates that there is a specific anti-biofilm activity of the compounds.

Both CBD and triclosan led to membrane permeabilization as shown by the appearance of a SYTO 9^low^ PI^high^ bacterial population, which was further increased upon drug combination. The increased membrane permeabilization goes along with the reduced bacterial viability. Moreover, the two compounds induced rapid transient membrane hyperpolarization that endured for a longer time when the two drugs were combined. Changes in membrane potential have implications for many vital processes in the bacteria [61] and are likely to contribute to the anti-bacterial activity of the combined drug treatment. The membrane hyperpolarization caused by triclosan has been related to its inhibitory action on the F_1_F_0_-ATPase required for pumping out protons [62]. The alteration in membrane permeability caused by the compounds might also contribute to the altered membrane potential. Triclosan has additionally been shown to inhibit glycolysis in *S. mutans* [62]. Likewise, the anti-bacterial action of CBD has been related to an interference in metabolic pathways [18]. Thus, the combined treatment of triclosan and CBD might concomitantly affect several biochemical processes, cumulating in an enhanced anti-bacterial response.

When inspecting the morphology of control and treated bacteria after a 2 h incubation, CBD led to an increase in the average cell length, and several electron-lucent rings resembling septa around the bacteria could be observed, suggesting impaired cell division. This provides a mechanistic insight into the bacteriostatic effect of CBD. An increase in the bacterial cell length was also observed with triclosan. Upon combination of these two drugs, the bacteria appear to be swollen, indicating that the bacteria are under stress and on the way to die.

The anti-biofilm activity of the combined CBD/triclosan treatment was shown by CV biomass staining, MTT metabolic activity, HR-SEM and SDCM. The HR-SEM images show an increase in EPS following the treatment of *S. mutans* with sub-MIC concentration of the single agents. This was also demonstrated by SDCM. However, the combined treatment resulted in strong reduction in bacterial adhesion to the surface. The few bacteria that still adhered to the surface showed a SYTO 9^low^ PI^high^ profile on SDCM, suggesting that they were mostly dead bacteria. Still, some EPS could be discerned under these conditions, which may have been produced by enzymes released from the dying bacteria. This indicates that the two compounds affect the viability of the bacteria and their ability to adhere to the surface, while the activity of the EPS-producing enzymes are still active. Since the EPS is still produced, although at a very low level, it is expected that combining triclosan and CBD with a glycosyltransferase inhibitor would further increase the efficiency in preventing *S. mutans* biofilm formation. Further studies are required to fully understand the mechanisms of how CBD and triclosan co-operate in exerting the anti-bacterial and anti-biofilm activities toward *S. mutans*.

## 5. Conclusions

The data presented in this paper show the mode of action of CBD/triclosan and the beneficial effect of the combined treatment against the oral cariogenic *S. mutans*. Taking into consideration the anti-inflammatory activities of CBD, besides its anti-bacterial and anti-biofilm activities, makes CBD a potential double-sword drug, and its incorporation into oral health care products together with triclosan might be advantageous for both preventing dental caries and oral inflammatory diseases.

## Figures and Tables

**Figure 1 biomedicines-11-00521-f001:**
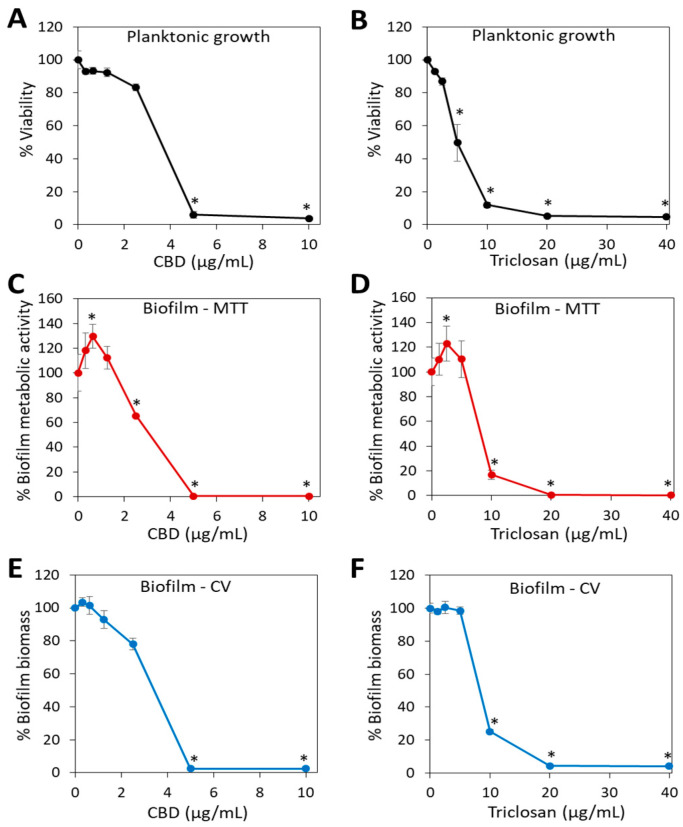
The anti-bacterial and anti-biofilm activity of CBD and triclosan against *S. mutans*. (**A**,**B**). *S. mutans* was exposed to various concentrations of CBD (**A**) or triclosan (**B**) in BHI for 24 h, and the planktonic growth was determined by measuring the OD at 600 nm. (**C**–**E**). *S. mutans* was exposed to various concentrations of CBD (**C**,**E**) or triclosan (**D**,**F**) in BHIS for 24 h, and the metabolic activity of the formed biofilms was measured by the MTT assay (**C**,**D**), and the biofilm biomass was determined by staining with crystal violet (CV) (**E**,**F**). * *p* < 0.05 when compared to control bacteria. *N* = 3.

**Figure 2 biomedicines-11-00521-f002:**
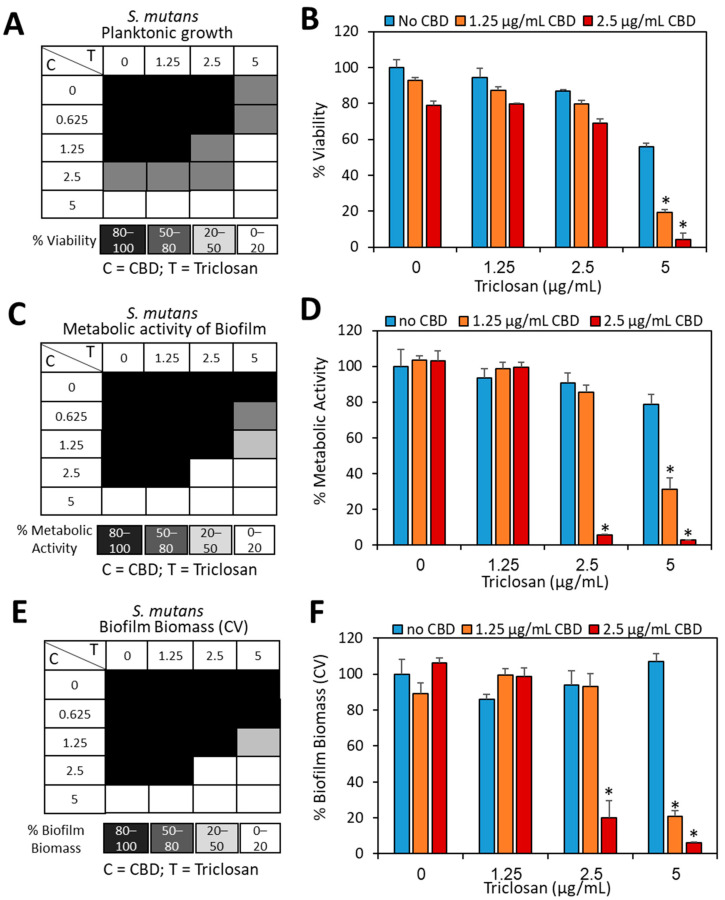
Increased anti-bacterial and anti-biofilm activity of combined triclosan/CBD treatment against *S. mutans*. (**A**). Checkerboard of triclosan and CBD on planktonic growth. (**B**). The % viability of *S. mutans* treated for 24 h with the indicated concentrations of triclosan and CBD. (**C**,**E**) Checkerboard of triclosan and CBD on biofilm formation as measured by MTT metabolic assay (**C**) or CV staining (**E**). (**D**,**F**) The % biofilm formation of *S. mutans* treated for 24 h with the indicated concentrations of triclosan and CBD as measured by MTT metabolic assay (**D**) or CV staining (**F**). * *p* < 0.05 in comparison to the treatment with each compound alone. *N* = 3.

**Figure 3 biomedicines-11-00521-f003:**
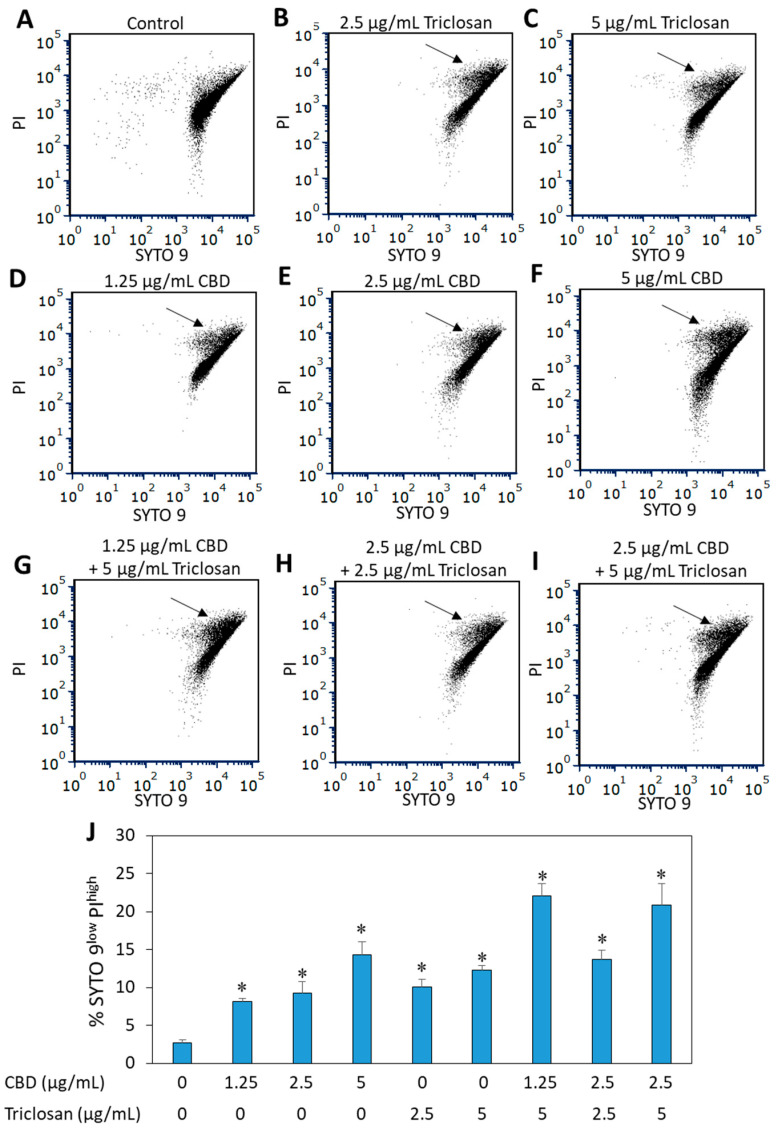
Increased membrane permeability following treatment with CBD and/or triclosan. (**A**–**I**) PI versus SYTO 9 dot plot analysis of *S. mutans* that was exposed to the indicated concentrations of triclosan and CBD for 1 h. The black arrows point to the SYTO 9^low^ PI^high^ bacterial population, which represents bacteria with disrupted membrane. (**J**) The percentage of the SYTO 9^low^ PI^high^ bacterial population. * *p* < 0.05 compared to control bacteria. *N* = 3.

**Figure 4 biomedicines-11-00521-f004:**
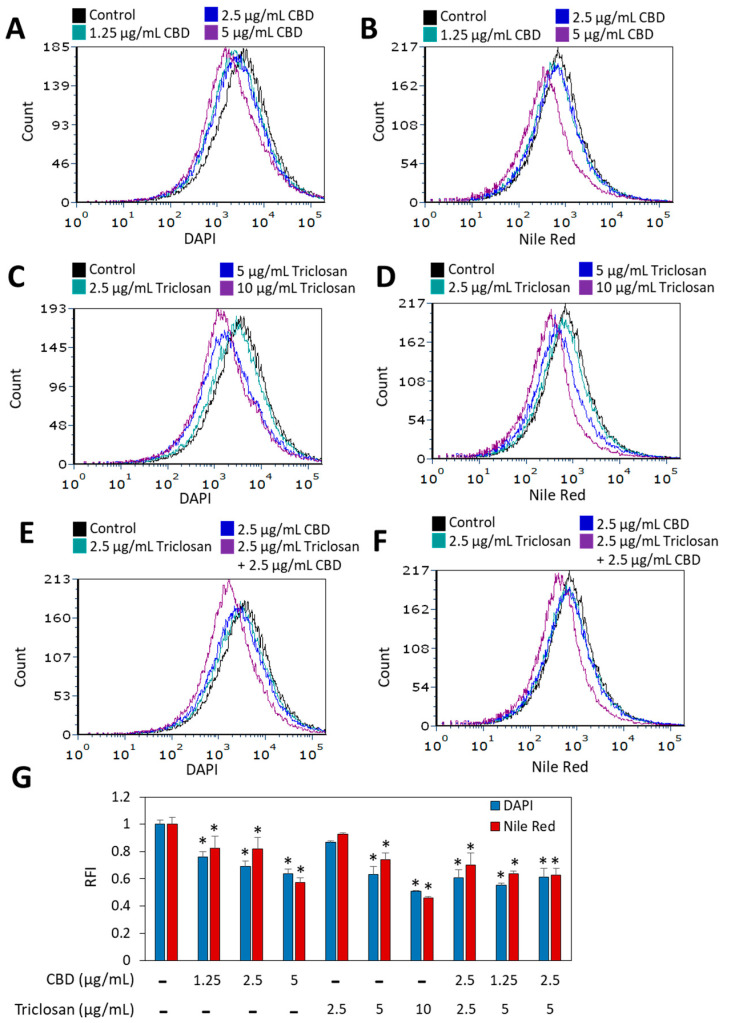
CBD and triclosan reduced both DAPI staining and Nile red staining of *S. mutans* after a 1 h incubation. (**A**–**F**) Flow cytometry of DAPI (**A**,**C**,**E**) and Nile Red (**B**,**D**,**F**) fluorescence intensities of *S. mutans* that was treated with the indicated concentrations of triclosan and CBD for 1 h. (**G**) The average relative fluorescence intensity (RFI) of DAPI and Nile Red in the control and treated bacteria. * *p* < 0.05 compared to control bacteria. *N* = 3.

**Figure 5 biomedicines-11-00521-f005:**
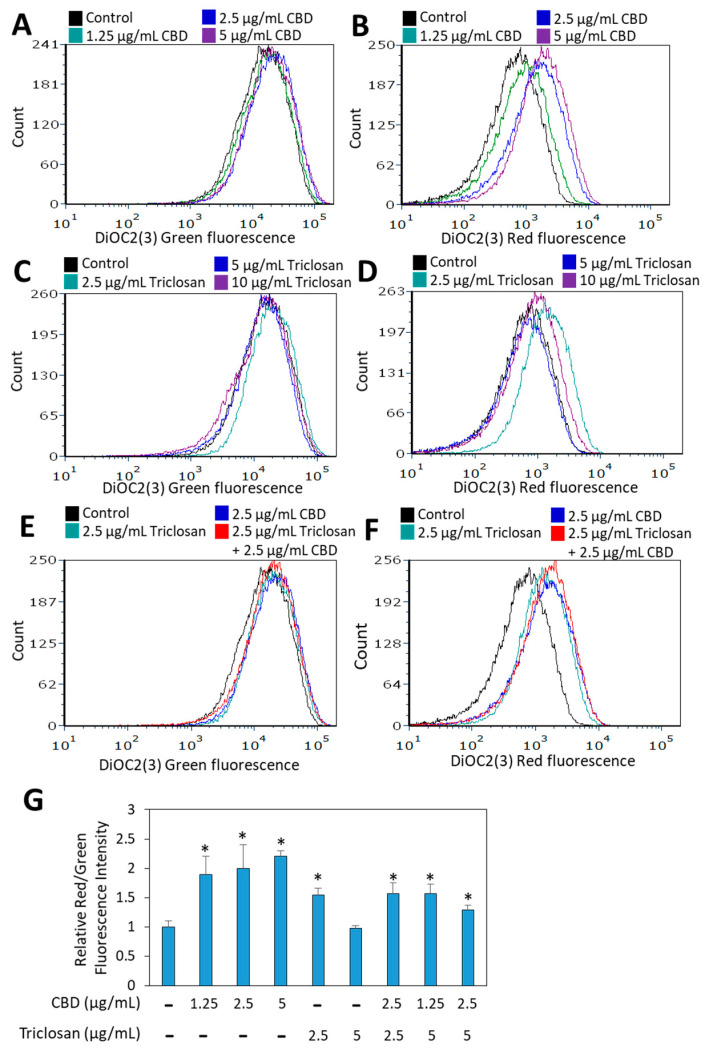
CBD and triclosan induced membrane hyperpolarization of *S. mutans* after a 1 h incubation. (**A**–**F**) Flow cytometry of DiOC2(3) green (**A**,**C**,**E**) and red (**B**,**D**,**F**) fluorescence intensities of *S. mutans* that was treated with the indicated concentrations of triclosan and CBD for 1 h. (**G**) The relative red/green fluorescence index of DiOC2(3) in the control and treated bacteria. An increase in this index indicates membrane hyperpolarization. * *p* < 0.05 compared to control bacteria. *N* = 3.

**Figure 6 biomedicines-11-00521-f006:**
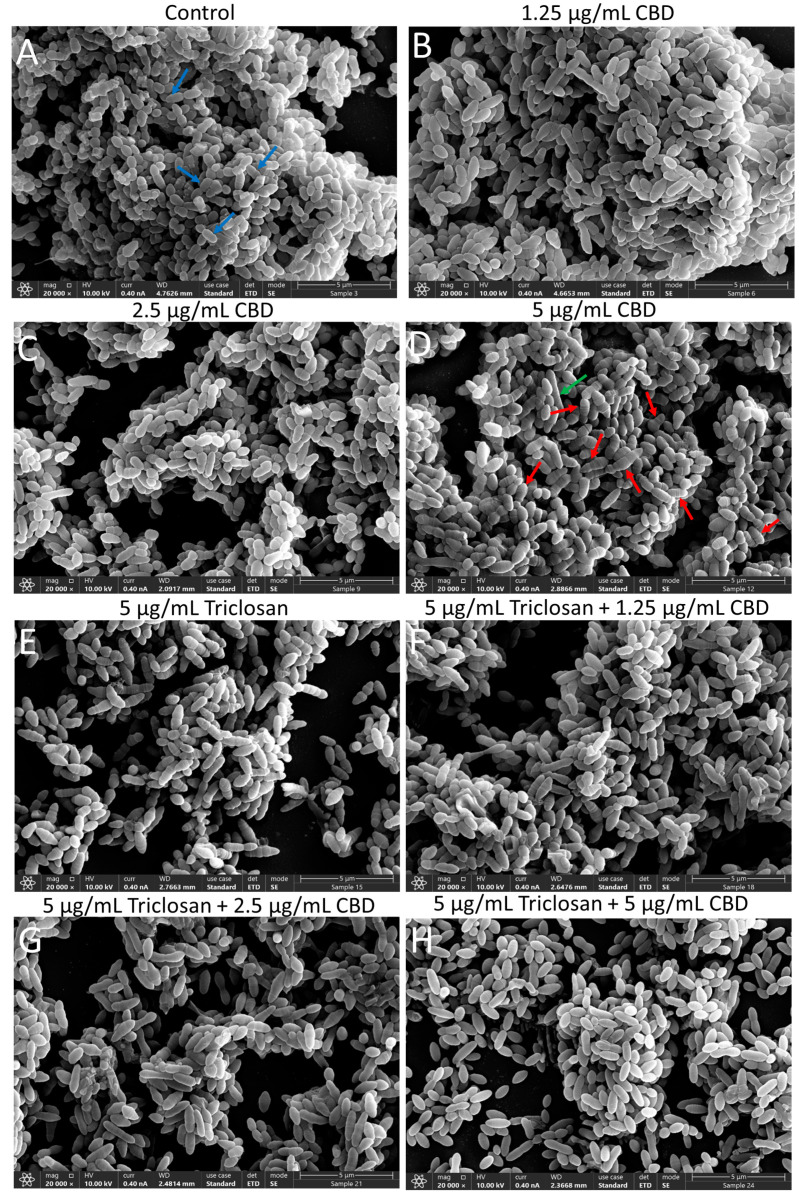
High-resolution scanning electron microscope (HR-SEM) images of planktonic growing *S. mutans* after a 2 h treatment with the indicated treatments (**A**–**H**). The blue arrows in (**A**) point to the classical ovococci form of *S. mutans*. The red arrows in (**D**) point to aberrantly shaped bacteria with multiple division septa. The green arrows point to extreme long bacteria. Uncropped images are presented in Appendix A.

**Figure 7 biomedicines-11-00521-f007:**
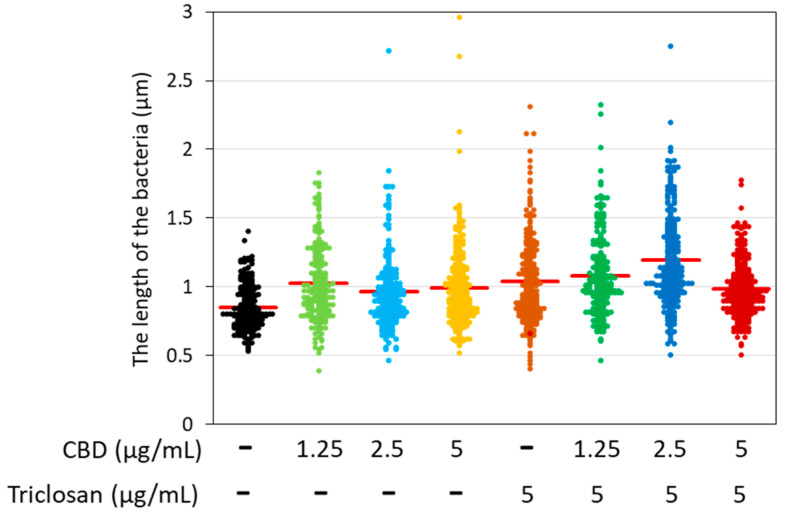
The length of the bacteria after a 2 h incubation with the indicated concentrations of triclosan and CBD as measured by ImageJ on 3–4 independent high-resolution scanning electron microscope (HR-SEM) images. *N* = 220–240 bacteria for each treatment. The red line shows the average bacteria length. The treated bacteria are significantly longer than the control bacteria (*p* < 0.05).

**Figure 8 biomedicines-11-00521-f008:**
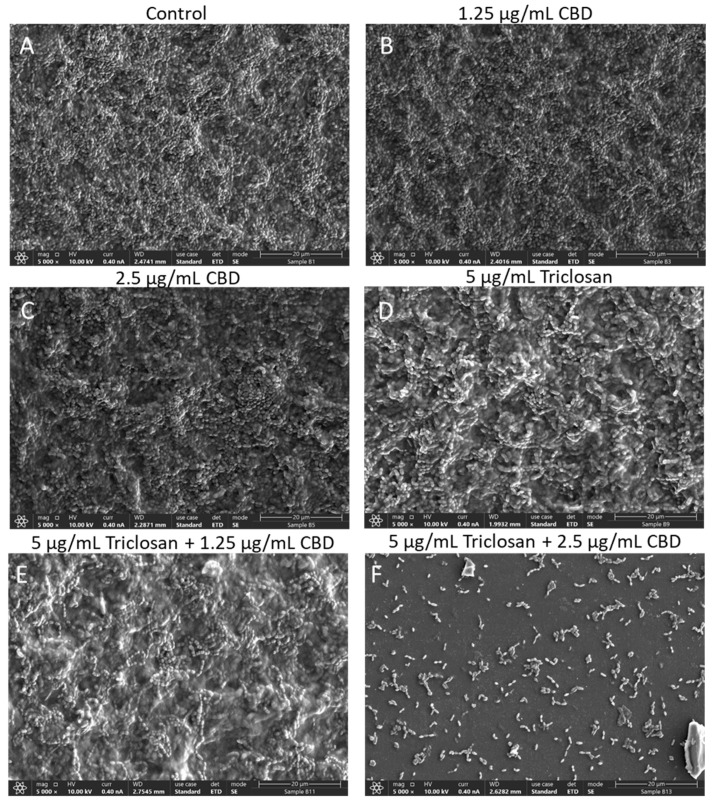
High-resolution scanning electron microscope (HR-SEM) images of *S. mutans* biofilms formed after a 24 h treatment with the indicated treatments. (**A**). Biofilm of control sample after a 24 h incubation in BHIS. (**B**–**E**). Biofilms formed in the presence of 1.25 μg/mL CBD (**B**), 2.5 μg/mL CBD (**C**), 2.5 μg/mL triclosan (**D**), 2.5 μg/mL triclosan with 1.25 μg/mL CBD (**E**) or 2.5 μg/mL triclosan with 2.5 μg/mL CBD (**F**) after a 24 h incubation in BHIS.

**Figure 9 biomedicines-11-00521-f009:**
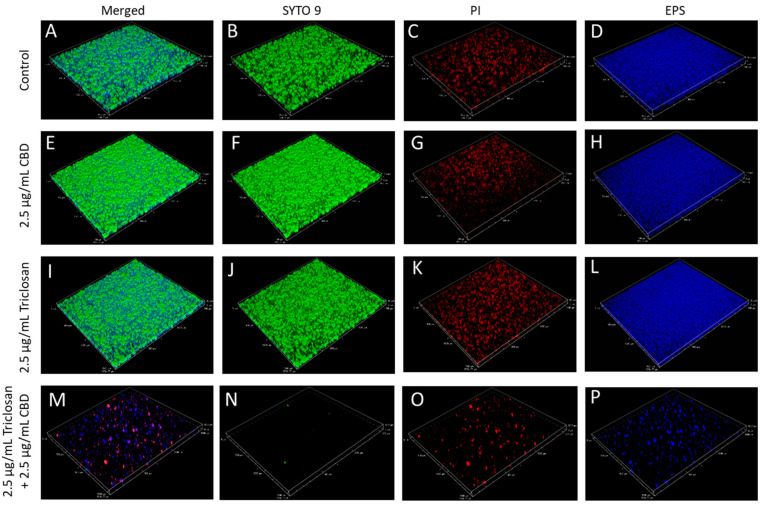
Spinning disk confocal microscope (SDCM) 3D images of *S. mutans* biofilms that were formed after a 24 h incubation with the indicated concentrations of triclosan and CBD. (**A**–**P**) The biofilms were stained with SYTO 9 (green fluorescence) (**B**,**F**,**J**,**N**), PI (red fluorescence) (**C**,**G**,**K**,**O**) and AlexaFluor^647^-conjugated Dextran 10,000 (presented in a blue color) (**D**,**H**,**L**,**P**), which stains EPS. The merged images are shown in (**A**,**E**,**I**,**M**). The combined treatment of triclosan with CBD results in the prevention of biofilm formation with only few dead bacteria attached to the surface.

**Figure 10 biomedicines-11-00521-f010:**
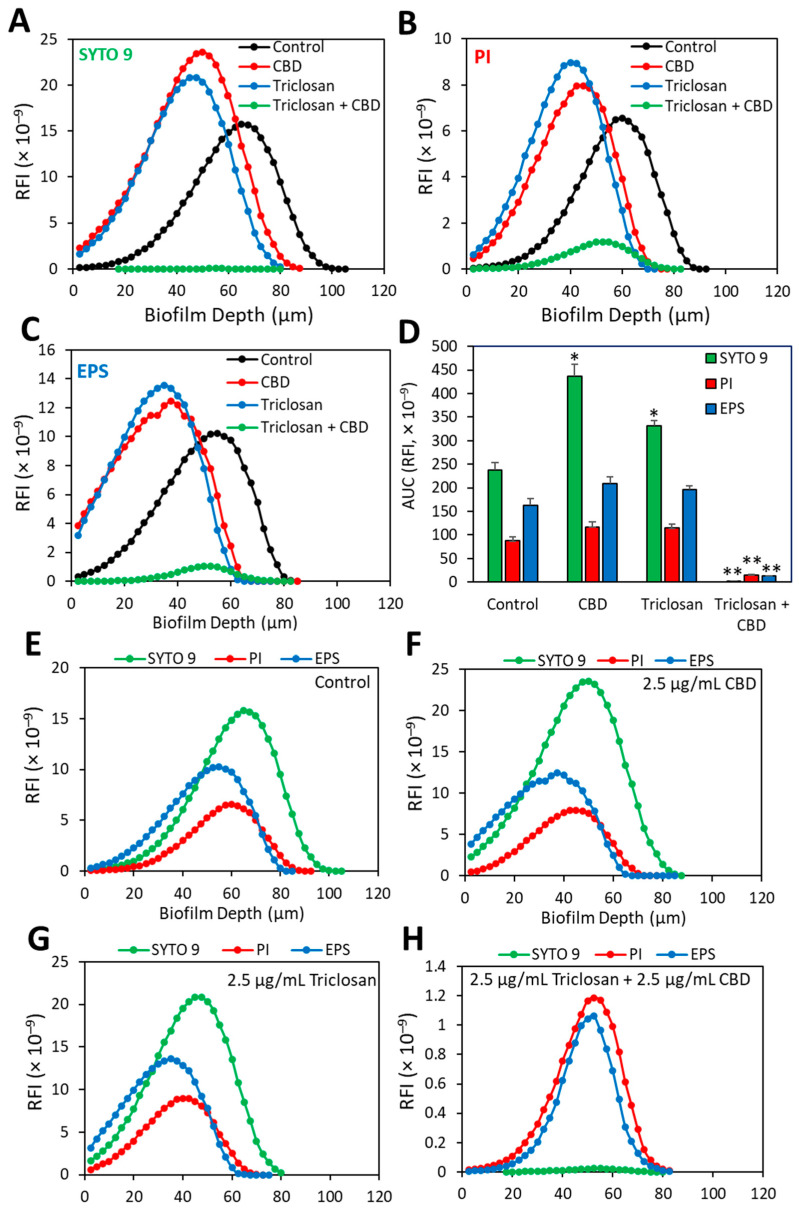
Quantification of the relative fluorescence intensities (RFIs) of SYTO 9, PI and fluorescent Dextran in the SDCM images of biofilms formed after a 24 h incubation with 2.5 µg/mL triclosan and/or 2.5 µg/mL CBD and control bacteria grown under the same conditions in the absence of the compounds. (**A**–**C**) The relative fluorescence intensity of SYTO 9 (**A**), PI (**B**) and fluorescent Dextran 10,000 (**C**) in control (black line), 2.5 µg/mL CBD (red line), 2.5 µg/mL triclosan (blue line) and 2.5 µg/mL triclosan + 2.5 µg/mL CBD-treated (green line) bacteria. (**D**) The average of the area under the curve (AUC) of the RFI in 4-5 different images of control or the treatment group. * *p* < 0.05 compared to control. ** *p* < 0.01 compared to control and the individual single treatments. (**E**–**H**) The relative fluorescence intensity of the three dyes in the four different groups: Control (**E**); 2.5 µg/mL CBD (**F**); 2.5 µg/mL triclosan (**G**) and 2.5 µg/mL triclosan + 2.5 µg/mL CBD (**H**). The red fluorescence of PI representing the dead bacteria become prominent after combined treatment of triclosan with CBD.

**Figure 11 biomedicines-11-00521-f011:**
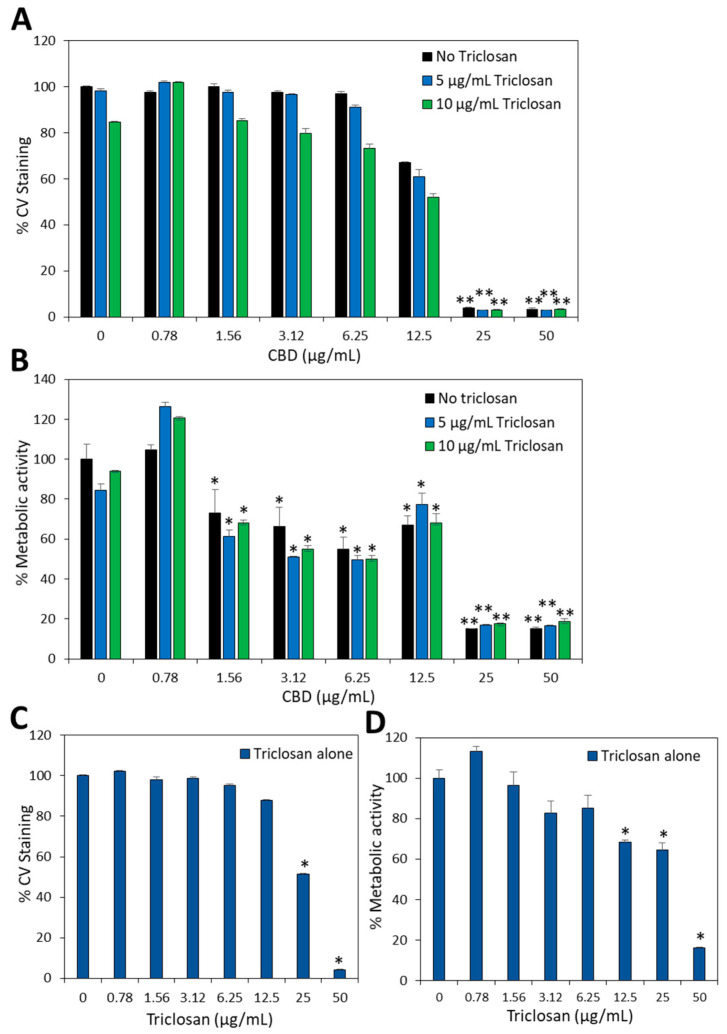
Biocompatibility assay on Vero epithelial cells. (**A**,**B**) Vero cell monolayer was incubated with various concentrations of CBD alone or in the presence of 5 or 10 µg/mL Triclosan for 24 h and then either stained with CV for measuring the total cell mass (**A**) or exposed to MTT to measure the metabolic activity (**B**). (**C**,**D**) Vero cell monolayer was incubated in the absence or presence of increasing concentrations of triclosan for 24 h and then either stained with CV for measuring the total cell mass (**C**) or exposed to MTT to measure the metabolic activity (**D**). * *p* < 0.05 when compared to control cells. ** *p* < 0.01 when compared to control cells. *N* = 3.

## Data Availability

Raw data are available upon reasonable request.

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
