# Peer review of "Improved Anti-Biofilm Effect against the Oral Cariogenic Streptococcus mutans by Combined Triclosan/CBD Treatment"

_biomedicines, 2023, doi:10.3390/biomedicines11020521_

Round 1

Reviewer 1 Report

The manuscript “Improved Anti-Biofilm Effect against the Oral Cariogenic Streptococcus mutans by Combined Triclosan/CBD Treatment” by Avraham et al. addressed strategies undertaken to prevent caries caused by Streptococcus mutans, specifically the beneficial role of combined triclosan and CBD treatment in potential caries protection. The Authors showed that increased biofilm formation was observed at sub-MIC concentrations of each compound alone while combining the drugs at these sub-MIC concentrations, the biofilm formation was prevented. Moreover, the concentrations required for the anti-bacterial and anti-biofilm activities toward S. mutans were non-toxic to the normal Vero epithelial cells. The manuscript is well-written and well-presented. The methodology at each stage of the study is well chosen, results are reliably demonstrated, and conclusions have application potential.

Concerns:

1.      In the Abstract lines 13-14, “Many strategies have been undertaken to prevent dental caries and periodontitis by targeting these bacteria.” S. mutans is not commonly considered a cause of periodontitis. It should be corrected.

2.      Line 117 “The morphology of the culture was verified under light microscopy (…)”

As I understand, this culture was liquid, so the morphology of the bacterial cells was verified under light microscopy, not the colony. This should be done on a solid medium to verify the morphology of S. mutans colony. Under the light microscope, all streptococci and staphylococci look the same. This should be clarified.

Author Response

Point-to-point Response
Reviewer 1
We thank the Reviewer for reading and critically reviewing our manuscript.

The manuscript “Improved Anti-Biofilm Effect against the Oral Cariogenic Streptococcus mutans by Combined Triclosan/CBD Treatment” by Avraham et al. addressed strategies undertaken to prevent caries caused by Streptococcus mutans, specifically the beneficial role of combined triclosan and CBD treatment in potential caries protection. The Authors showed that increased biofilm formation was observed at sub-MIC concentrations of each compound alone while combining the drugs at these sub-MIC concentrations, the biofilm formation was prevented. Moreover, the concentrations required for the anti-bacterial and anti-biofilm activities toward S. mutans were non-toxic to the normal Vero epithelial cells. The manuscript is well-written and well-presented. The methodology at each stage of the study is well chosen, results 
are reliably demonstrated, and conclusions have application potential.
Concerns:
Issue 1: In the Abstract lines 13-14, “Many strategies have been undertaken to prevent dental caries and periodontitis by targeting these bacteria.” S. mutans is not commonly considered a cause of periodontitis. It should be corrected.
Answer: Thank you for the comment. We have now removed “and periodontitis” from the Abstract.
Issue 2. Line 117 “The morphology of the culture was verified under light microscopy (…)”
As I understand, this culture was liquid, so the morphology of the bacterial cells was verified under light microscopy, not the colony. This should be done on a solid medium to verify the morphology of S. mutans colony. Under the light microscope, all streptococci and staphylococci look the same. This should be clarified.
Answer: Streptococcus mutans shows a characteristic coccoid-to-ellipsoid morphology and forms chains which differ for the Staphylococcus aureus that form round, spherical diplococci or “grape-like” clusters. We also have tested the purity of the bacteria by seeding them on BHI-agar plates, where they form characteristic minute colonies. We have according to the suggestion by 
the reviewer modified the text to: “The chain formation and coccoid-to-ellipsoid morphology of the S. mutans culture were verified under light microscopy (Axio Lab.A1, Carl Zeiss Jena GmbH, Germany) using the x100 lens…” and added the following text: ”The purity of the culture was tested by seeding the bacteria on BHI-agar plates where they formed distinctive minute colonies that are strongly attached to the agar.”

Reviewer 2 Report

In this work, the authors have studied the beneficial role of combined triclosan/CBD treatment for potential protection against caries. It is a good and complete work and I recommend its publication. 

Author Response

We thank the Reviewer for reading and critically reviewing our manuscript and for his comments.

No revision was required.

Reviewer 3 Report

A manuscript titled "Improved Anti-Biofilm Effect against the Oral Cariogenic Streptococcus mutans by Combined Triclosan/CBD Treatment" presents interesting content that is part of the current trend of searching for new antibacterial drugs, limiting the antimicrobial resistance and also ongoing research on application of Cannabis sativa in medicine.

The authors present this issue in relation to Streptococcus mutans (planktonic and biofilm forms) responsible for oral diseases and show the potential of cannabidiol to enhance the already existing prophylaxis. I think that these are sufficient reasons for the work to be of great interest to readers of various specialties.

From the methodological point of view, where modern research tools were used, the work is described in such details that it leaves no doubts for a microbiologist researcher. The results are also presented in an understandable, clear way. However, the discussion, in my opinion, leaves a certain dissatisfaction. After the extensive introduction, we have a small part of the discussion on the results obtained.

Other, minor remarks are as follows:

1. L. 44: what does "facultative coccus" mean? In my opinion it should be "facultatively anaerobic coccus"

2. L. 127: what exactly type of controls were used in this experiment? It should be described.

3. Figure 1: I would not combine plankton and biofilm data determind with different methods in one graph. For readers this causes difficulties in the quick analysis, at the beginning of the article. I suggest dividing this data into 6 separate small graphs (maybe added to the supplementary figures).

4. The arrows in the SEM images should be in bold because they are barely visible.

5. SEM images at this magnification do not show any significant difference. Although the enlargements are added in supplementary Fig 3, I would suggest incorporating magnification of the morphologically changed cells to the already existing images, in Fig 6.

Author Response

Response to Reviewer 3:

We thank the Reviewer for reading and critically reviewing our manuscript.

A manuscript titled "Improved Anti-Biofilm Effect against the Oral Cariogenic Streptococcus mutans by Combined Triclosan/CBD Treatment" presents interesting content that is part of the current trend of searching for new antibacterial drugs, limiting the antimicrobial resistance and also ongoing research on application of Cannabis sativa in medicine.

The authors present this issue in relation to Streptococcus mutans (planktonic and biofilm forms) responsible for oral diseases and show the potential of cannabidiol to enhance the already existing prophylaxis. I think that these are sufficient reasons for the work to be of great interest to readers of various specialties.

From the methodological point of view, where modern research tools were used, the work is described in such details that it leaves no doubts for a microbiologist researcher. The results are also presented in an understandable, clear way.

Issue 1: However, the discussion, in my opinion, leaves a certain dissatisfaction. After the extensive introduction, we have a small part of the discussion on the results obtained.

Answer: We have now added text to the discussion.

Other issues:  minor remarks are as follows:

  1. L. 44: what does "facultative coccus" mean? In my opinion it should be "facultatively anaerobic coccus"

Answer: Thank you for the comment. We have now corrected the text to “facultatively anaerobic coccus”.

  1. L. 127: what exactly type of controls were used in this experiment? It should be described.

Answer: In the text we wrote “Control samples got the same concentrations of ethanol as the treated samples.” In order to clarify the issue, we have added the following text: “In parallel, untreated bacteria were incubated in BHI. The presence of ethanol at the concentrations used (0.0156-1%) had no effect on bacterial growth.”

  1. Figure 1: I would not combine plankton and biofilm data determind with different methods in one graph. For readers this causes difficulties in the quick analysis, at the beginning of the article. I suggest dividing this data into 6 separate small graphs (maybe added to the supplementary figures).

Answer: According to the Reviewer’s recommendation we have divided Figure 1A and B into 6 subfigures.

  1. The arrows in the SEM images should be in bold because they are barely visible.

Answer: According to the Reviewer’s recommendation, we have made the arrows in bold.

  1. SEM images at this magnification do not show any significant difference. Although the enlargements are added in supplementary Fig 3, I would suggest incorporating magnification of the morphologically changed cells to the already existing images, in Fig 6.

Answer: Since all of the figure should fit into one page, we cannot expand the figure to 8 pages. Therefore, the full-sized images are presented in the Supplementary data (Supplementary Figure 3). For the interested readers, we have now added all of the images of Figure 6 in Supplementary Figure 3 in full size.
